# Recovery from recurrent depression with mindfulness-based cognitive therapy and antidepressants: a qualitative study with illustrative case studies

Alice Tickell,[1] Richard Byng,[2] Catherine Crane,[1] Felix Gradinger,[2] Rachel Hayes,[1] James Robson,[3] Jessica Cardy,[1] Alice Weaver,[1] Nicola Morant  ,[4] Willem Kuyken  [1]

[1]Department of Psychiatry, University of Oxford, Oxford, UK
[2]Peninsula Medical School, Faculty of Health, University of Plymouth, Plymouth, UK
[3]Department of Education, University of Oxford, Oxford, UK
[4]Department of Psychiatry, University College London, London, UK

**Correspondence to**
Dr Willem Kuyken;
willem.kuyken@psych.ox.ac.uk

## ABSTRACT

**Objectives** This study aimed to describe the recovery journeys of people with a history of recurrent depression who took part in a psychosocial programme designed to teach skills to prevent depressive relapse (mindfulness-based cognitive therapy (MBCT)), alongside maintenance antidepressant medication (ADM).

**Design** A qualitative study embedded within a multicentre, single blind, randomised controlled trial (the PREVENT trial).

**Setting** Primary care urban and rural settings in the UK.

**Participants** 42 people who participated in the MBCT arm of the parent trial were purposively sampled to represent a range of recovery journeys.

**Interventions** MBCT involves eight weekly group sessions, with four refresher sessions offered in the year following the end of the programme. It was adapted to offer bespoke support around ADM tapering and discontinuation.

**Methods** Written feedback and structured in-depth interviews were collected in the 2 years after participants undertook MBCT. Data were analysed using thematic analysis and case studies constructed to illustrate the findings.

**Results** People with recurrent depression have unique recovery journeys that shape and are shaped by their pharmacological and psychological treatment choices. Their journeys typically include several over-arching themes: (1) beliefs about the causes of depression, both biological and psychosocial; (2) personal agency, including expectations about their role in recovery and treatment; (3) acceptance, both of depression itself and the recovery journey; (4) quality of life; (5) experiences and perspectives on ADM and ADM tapering-discontinuation; and (6) the role of general practitioners, both positive and negative.

**Conclusions** People with recurrent depression describe unique, complex recovery journeys shaped by their experiences of depression, treatment and interactions with health professionals. Understanding how several themes coalesce for each individual can both support their recovery and treatment choices as well as health

## Strengths and limitations of this study

► Recurrent depression is a leading cause of disability adjusted life years and antidepressant medication (ADM) is the mainstay approach to treatment; this study is the first to describe people's experiences of recovery with an ADM alongside a psychosocial approach designed to support recovery (mindfulness-based cognitive therapy (MBCT)).
► The sample was relatively large and purposively sampled to illustrate a range of recovery journeys and outcomes.
► Participants experiences in the 2 years following MBCT were sampled, using an innovative approach to supporting participants' to describe the richness of their experiences of recovery and treatment.
► The sampling necessarily meant that we did not include people with a history of recurrent depression who had decided against ADM, tapering and discontinuing their ADM and/or a psychological approach.

professionals in providing more accessible, collaborative, individualised and empowering care.

**Trial registration number** Clinical trial number ISRCTN26666654; post results.

Depression is a major public health problem. Globally more than 264 million people suffer from depression, and lifetime prevalence rates are estimated to be between 6% and 20%.[1 2] Furthermore, depression can be a relapsing and recurring condition, and on average people who experience one episode of depression have seven or eight episodes over their lifetime.[3] Clinical guidance typically recommends that people with recurrent depression should take maintenance antidepressant medications (ADMs) after remission or engage in preventive psychological

interventions to maintain recovery.[4] For people with recurrent depression, their recovery journey is shaped by their experiences of depression, 'illness model', interactions with mental health professionals, treatments that have and have not worked and expectations about what recovery will entail.[5] For many, ADMs are an important part of their recovery. Reasons for long-term ADM use include positive experiences of ADMs, fear of relapse, perceived lack of alternatives and concerns about withdrawal effects.[6] On the other hand, people also describe a number of reasons for wanting to discontinue ADMs, including feeling better and wanting to test whether depression has gone away, ambivalence and uncertainty about the role of ADMs in recovery, side effects outweighing benefits, questioning whether the self on ADMs is the 'real self', and wanting to assert control over their well-being.[7]

Research and clinical guidelines suggests that psychological therapies, such as mindfulness-based cognitive therapy (MBCT) and cognitive-behavioural therapy (CBT), can support recovery from depression as well as support discontinuing ADMs.[4 8–10] A significant proportion of people express a preference for psychological therapies, so they can learn strategies that support recovery without the need for long-term reliance on ADMs.[11] But psychological therapies can be difficult to access, involve significant investment of time and energy and are not effective for everyone.[12]

While studies have examined people's experiences of ADM and MBCT in recovery, none have focused primarily on how they operate alongside one another. This study explores how people with a history of recurrent depression describe their experience of using MBCT and ADM to support their recovery, drawing on both written feedback booklets and more in-depth interviews. It was embedded within a randomised controlled trial (RCT) comparing MBCT with support to taper and discontinue ADMs (henceforth MBCT-TS) and maintenance antidepressants over a 2-year period.[13] These findings could inform decision-making between general practitioners (GPs) and patients about the journey of management and recovery from recurrent depression.

## METHODS
### Study context
This qualitative process evaluation was embedded within a randomized controlled trial comparing mindfulness-based cognitive therapy with maintenance anti-depressant treatment in the prevention of depressive relapse/recurrence (the PREVENT trial). This was a multicentre, single blind, parallel RCT, which investigated whether MBCT-TS (n=212) was superior to maintenance ADMs (n=212) for the prevention of depressive relapse or recurrence over 24 months (trial design is described by Kuyken et al).[13–16] The trial found that MBCT-TS was not superior to maintenance ADM in preventing depressive relapse over 2 years; however, a subsequent individual patient data

meta-analysis which included this data suggests MBCT as an alternative to maintenance ADMs.[8] We present a statement concerning reflexivity in the online supplementary materials, which outlines the experience and background of the authors, to acknowledge our theoretical positions and values in relation to this study (see online supplementary 1).[17] We also include the consolidated criteria for reporting qualitative research (COREQ) checklist to match our procedures against standard criteria for qualitative research (see online supplementary 2).

### Participants
Participants in the PREVENT trial were recruited from 95 general practices in urban and rural settings in four UK centres, in addition to self-referral.[13] Inclusion criteria were a diagnosis of recurrent major depressive disorder in full or partial remission according to the Diagnostic and Statistical Manual of Mental Disorders-IV[18]; three or more previous major depressive episodes; age 18 years or older; and on a therapeutic dose of maintenance antidepressant drugs in line with the British National Formulary and NICE (National Institute for Health and Care Excellence) guidance.[4] Exclusion criteria were a current major depressive episode, comorbid diagnoses of current substance misuse; organic brain damage; current or past psychosis, including bipolar disorder; persistent antisocial behaviour; persistent self-injury needing clinical management or therapy; and formal concurrent psychotherapy. All participants gave informed consent before participating in the trial. The full process of recruitment for the PREVENT trial is described in Kuyken et al.[16]

This study examined a sub-group of participants from the PREVENT trial (n=42) allocated to the MBCT-TS arm of the trial. Of the 212 participants allocated to receive MBCT-TS, 176 received an adequate dose of treatment (attended four or more group sessions of therapy).[13] The researchers purposively sampled a sub-group of these participants (n=46) to represent a spread of characteristics and experiences with respect to: whether they reported their childhood as having higher or lower levels of abuse, treatment response (relapse/no relapse to a major depressive episode), and ADM discontinuation profile across the 24-month follow-up period (discontinued ADMs, discontinued ADMs but subsequently resumed them, tapered ADMs but never fully discontinued, never tapered or discontinued ADMs).[13] Of the 46 people invited to interview, 42 agreed, which comprised the final sample. Of the four who declined, two had moved away from the area, one was not interested in participating and one participant had changed their contact details and could not be reached. Interviewees did not differ in either baseline characteristics or trial outcomes from the broader study sample (see table 1).

### MBCT-TS intervention
MBCT-TS comprised MBCT delivered in line with the published treatment manual,[19] but adapted to include a greater focus on developing a relapse/recurrence

**Table 1** Characteristics of the sample

| | Interviewed (n=42) | All MBCT-TS participants (n=212) |
|---|---|---|
| **Demographic characteristics** | | |
| Female (%) | 31 (74) | 151 (71) |
| White (%) | 42 (100) | 210 (99) |
| Age (in years) | | |
| M (SD) | 51.88 (10.51) | 50 (12) |
| Range | 25–72 | 22–78 |
| **Psychiatric characteristics** | | |
| Previous episodes | | |
| <6 episodes | 26 (62) | 120 (57) |
| ≥6 episodes | 16 (38) | 92 (43) |
| Comorbid mental health diagnoses | | |
| 1 or more (%) | 15 (36) | 75 (35) |
| **Treatment preference at baseline** | | |
| MBCT-TS preference (%) | 34 (81) | 150 (71) |
| ADM preference (%) | 1 (2) | 12 (6) |
| No preference (%) | 7 (17) | 50 (24) |
| **Treatment outcome** | | |
| Relapse | | |
| N (%) that relapsed during the follow-up phase | 23 (55) | 94 (44) |
| Antidepressant usage during the follow-up phase | | |
| Stopped and stayed stopped (%) | 13 (31) | 67 (32) |
| Stopped and resumed (%) | 9 (21) | 57 (27) |
| Reduced but never stopped (%) | 9 (21) | 29 (14) |
| Never stopped or reduced (%) | 11 (26) | 23 (11) |
| Residual depression symptoms | | |
| BDI score at baseline, M (SD) | 15.90 (11.35) | 13.8 (12.4) |
| BDI score at 24-month follow-up, M (SD) | 12.39 (12.25) | 11.6 (10.9) |

ADM, antidepressant medication; BDI, Beck Depression Inventory; MBCT-TS, mindfulness-based cognitive therapy with support to taper and discontinue ADMs.

signature and response plan that explicitly included participants' reduction/discontinuation of ADM (see Kuyken et al[16] for more detail). The programme involved eight 2¼-hour group sessions, normally over consecutive weeks, with up to four refresher sessions offered in the year following the end of the 8-week programme. Researchers encouraged participants in the MBCT-TS arm to taper and discontinue their maintenance ADMs at several points from the middle of the MBCT-TS course onwards, and provided information to GPs and participants about typical tapering/discontinuation regimes and possible withdrawal effects. If participants experienced a relapse/

recurrence during the course of the trial, researchers encouraged them to discuss the most appropriate treatment with their GP and made no further requests that they consider tapering/discontinuing their ADMs.

### Qualitative data collection
We used both written feedback booklets (collected at two time points, soon after MBCT and then at study end) and interviews (at study end) to gather participants' more in-depth experiences of recovery. We combined each participant's interview and written feedback booklet data to form a single account of their experiences. This formed the study's data corpus.

### Written feedback booklets
One month after completing MBCT-TS all trial participants were invited to complete a feedback booklet addressing attitudes towards, and experiences of, taking and reducing antidepressant medication; experiences of taking part in MBCT-TS and MBCT-TS practices; and the impact of MBCT. In addition to the earlier points, participants received a further feedback booklet 24 months later, which asked the same questions as the first booklet but also included questions focused on participants' experiences in the follow-up period and basic data on the amount and type of mindfulness practice. The booklets are provided in the online supplementary materials (see online supplementary 3 and 4).

### Interviews
Interviews were semi-structured and normally conducted face-to-face by trained researchers, approximately 24 months after MBCT-TS. They lasted between 45 min and 1 hour and explored experiences during the follow-up period, with questions addressing times of wellness, early signs of potential depressive relapse and relapses. Questions explored the use and perceived value of mindfulness techniques, use of ADMs and their combination. We tailored interviews to the specific profile of each participant using a 'timeline' prepared in advance and amended by the participant at the interview, which summarised each participant's ADM use, relapses and significant life events, as reported to the research team during the trial. The interview schedule was deliberately broad in focus (see online supplementary 5). Interviews were recorded and transcribed for analysis.

### Public and patient involvement
The PREVENT trial benefited from the expertise of many people with lived experience of mental health difficulties including a number of members of a locally organised voluntary group called the lived experience group (LEG). The LEG assisted the PREVENT trial at every stage of its development including both the interview schedule and written feedback booklets. There were reviewed and then trialled by several members of the LEG who suggested a number of fundamental changes. A member of the LEG also provided specific training to the research staff on conducting interviews.

## Data analyses

We used thematic analysis as our analytic approach.[17] First, we selected eight participants with a range of ADM discontinuation journeys during the trial period: two who had discontinued ADMs and remained ADM-free; two who had discontinued ADMs and subsequently resumed; two who had never tapered or discontinued ADMs; and two who had tapered but never discontinued ADMs. Four researchers (AT, CC, JR and WK) independently analysed the interview transcripts and accompanying 1-month and 24-month feedback booklets for each participant. In this phase, we conducted inductive analysis, with each researcher developing a preliminary coding frame. These frames were then integrated through discussion to remove redundancies and ensure breadth. This collaboratively produced, inductive coding frame was then combined with deductive codes developed from key literature on participant experiences of MBCT,[20 21] and ADM use,[7 22] to establish a working coding frame.

The lead researcher (AT) then analysed the 42 interviews and accompanying booklets against this coding frame, using NVivo V.11 software. AT, CC and JR met at regular intervals to discuss additional emerging codes and arising themes and, if deemed appropriate, integrated these into the coding frame. Midway through coding, AT sought peer feedback on emerging themes from co-authors, at an internal research meeting and at a symposium focused on antidepressant tapering at an international conference (Tickell, 2018). Feedback from these presentations helped clarify which themes were particularly important, and in particular helped the researchers reflect on those that related specifically to participants' experiences of ADM alongside MBCT-TS. Once the data were fully coded, the researchers reviewed the themes in the light of the core research question. These were discussed with the wider authorship group, whose input was used to reduce redundancy across themes, and highlight their interactions. Peer review of an earlier version of this manuscript also led to further refining the questions and themes. Finally, we identified cases that illustrated the unique stories of recovery and the ways the common themes coalesced in different ways in illustrative case studies.

## RESULTS

The thematic analysis yielded six over-arching themes, each with a number of constituent sub-themes. We provide a summary (table 2) and narrative account of each theme and its constituent sub-themes, illustrating these with extracts from participants' accounts. While each person's experience of MBCT and ADMs was unique, these themes converged in complex ways within individual case. Five case examples illustrate these different recovery journeys (box 1).

**Table 2** Themes and sub-themes

| Theme | Sub-themes |
|---|---|
| **Beliefs about the causes of depression** | *Neurochemical disruption* *Learning a psychological model* *Integrating models* |
| **Personal agency** | *Control over depression* *Responsibility* |
| **Acceptance** | *Resolving shame* *Self-care* *Perspectives on relapse* |
| **Quality of life** | *Experiencing emotions more fully* *From coping to enjoying life* |
| **ADM tapering/discontinuation** | *Pace of reduction* *Managing withdrawal effects* |
| **Interactions with GP** | *Presence and support* *Following advice* |

ADM, antidepressant medication; GP, general practitioners.

### Beliefs about the causes and treatment of depression

This over-arching theme describes participants' beliefs about the causes of depression and how these beliefs influence their treatment decisions. This theme comprises three sub-themes.

#### Neurochemical disruption

Many participants described entering the study believing that their recurrent depression was due to a neurochemical disruption in their brain, often citing specifically a deficiency or imbalance of the neurotransmitter serotonin. Participants viewed medication as a way to correct this issue and made parallels to biomedical disorders, viewing ADMs as a 'physiological need' in the same way that 'diabetics require insulin' because 'there is some chemical missing' *(2102; Written feedback, Never tapered or discontinued)*. For instance, Annie explained that she went on ADMs because her doctor told her that she had lower levels of serotonin than other people (see box 1). This belief appeared to influence expectations about psychological therapy, as some participants stated that they did not understand how 'mindfulness would be able to counteract depression […] if it's generated by a chemical imbalance' *(1031, Interview, Never tapered or discontinued)*. Other people said that they had not given much thought to why they were depressed or how ADMs worked: 'Happy pills […] I've never really given it a great deal of thought exactly what they do to be honest. […] I just know I don't feel so bad with them' *(2123, Interview, Tapered but never discontinued)*.

#### Learning a psychological model

Participants described how their views on the causes of depression evolved during and following the MBCT-TS

**Box 1   Case examples**

**Mandy**. *Mandy, aged 57, experienced nine episodes of depression, beginning when she was 32. Following the mindfulness-based cognitive therapy with support to taper and discontinue ADMs (MBCT-TS) course Mandy successfully discontinued her antidepressant medication (ADM) treatment. She did not experience a relapse over the 24-month follow-up period.*

Mandy recalled how ADMs had helped her to function well. In the past, she had tried tapering, but had always relapsed, so assumed that ADMs would be a part of her life forever. At first, Mandy was nervous, but was willing to try tapering ADMs gradually and with the support of MBCT-TS (**Personal Agency: Control Over Depression**). Mandy's GP was supportive, but reassured her that it was ultimately her decision (**Interactions with GP: Presence and Support**). During the MBCT-TS course, Mandy said that she learned a different model of depression and developed a better understanding of 'how the mind works' (**Beliefs about the Causes of Depression: Learning a Psychological Model**). She felt more confident about tapering, and said that this time it was 'so easy, knowing that I have been given tools to help me through it if needed', and found the course 'totally liberating' as it gave her the chance to take control of her depression, rather than the other way round (**Personal Agency: Responsibility**). She also found it helpful to learn about the possible symptoms of withdrawal, which included mood swings. Mandy realised that the relapses she had experienced when she had tried to taper her ADMs in the past might have been withdrawal symptoms, as opposed to 'real relapses' (**ADM Tapering / Discontinuation: Managing Withdrawal Effects**). At the time of interview, having discontinued ADMs, Mandy still practised what she learned in MBCT-TS and made it part of her daily routine. She accepted that if she ever relapsed, she could use ADMs, but it would only ever be a short-term solution, because she has the MBCT-TS skills as a 'weapon' to help her manage (**Acceptance: Self-Care**).

**Greta**. *Greta, aged 72, had experienced three episodes of depression, beginning when she was 33. Following the MBCT-TS course she discontinued her ADMs but then resumed following a deterioration in mood.*

Greta was very optimistic about the course because she hated being on ADMs, which gave her unpleasant side effects that interfered with her quality of life (**Personal Agency: Control over Depression**). At first, Greta said the course made an 'immense difference' to her, and she described learning how to combat the negative thoughts and feelings she was having (**Beliefs about the Causes of Depression: Learning a Psychological Model**). The programme left Greta feeling 'so well and positive' that she decided to taper her ADMs very quickly (**ADM Tapering/Discontinuation: Pace of Reduction**), but began to feel her mood dipping. Greta thought this must be a sign that the programme was not working, because she should not feel depressed (**Acceptance: Perspectives on Relapse**). Greta went to her GP, who did not seem interested in the programme and told her to resume ADMs immediately (**Interactions with GP: Presence and Support**). She was disappointed and felt 'guilty' that she was not able to use these new skills to keep herself well (**Personal Agency: Responsibility**). She stopped practising mindfulness, although the programme made her remember to appreciate the high points in her day and experience more joy (**Quality of Life: From Coping to Enjoying Life**).

**Annie**. *Annie, aged 48, had experienced five episodes of depression, beginning when she was 23. Following the MBCT-TS programme, she discontinued her ADMs but then resumed them later.*

Annie felt that ADMs had a positive impact on her life, allowing her to cope day-to-day as a full-time carer for her husband who had a disability. At first, she was very reluctant to try discontinuing ADMs because

she believed she might have low levels of serotonin (**Beliefs about the Causes of Depression: Neurochemical Disruption**). However, the programme taught her a new model of understanding depression (**Beliefs about the Causes of Depression: Learning a Psychological Model**), which made her feel empowered to practice the psychological techniques (**Personal Agency: Responsibility**). She started to taper off ADMs, but then her mother died and her husband's health deteriorated, so it was difficult to find time to practice mindfulness. Her GP advised her it was probably not a good time to discontinue (**Interactions with GP: Presence and Support**), so she resumed ADMs. However, Annie still incorporated the mindfulness exercises into her everyday life, which brought her more joy (**Quality of Life: Experiencing Emotions More Fully**). She also recognised that it is not her fault when she felt depressed, given how challenging her life was (**Acceptance: Resolving Shame**). Annie felt that the best way to manage her depression was to combine ADMs with mindfulness practices, which gave her more skills to look after herself during difficult times (**Acceptance: Self-Care**). She felt hopeful that 1 day she would discontinue ADMs, when her life circumstances were more stable.

**George**. *George, aged 37, had experienced ten episodes of depression, beginning when he was 16. Following the MBCT-TS programme he discontinued his ADMs. He experienced a relapse to depression during the 24-month follow-up.*

George was very optimistic about trying an alternative to ADMs, because they made him feel like a 'zombie'. Having experienced substance misuse issues in the past George had the goal of being totally 'chemical free'. Before the course, George felt he had no control over his depression symptoms, and his mood would deteriorate suddenly without warning (**Personal Agency: Control**). Through practising the mindfulness skills, he described developing more awareness of his emotions and felt he would have the skills to manage them (**Personal Agency: Responsibility**). George said that the best part of taking part in MBCT-TS was meeting other people with depression, which made him feel more accepting of himself (**Acceptance: Resolving Shame**). He felt that ADMs had masked his symptoms, whereas MBCT-TS allowed him to explore the problems in his life that were contributing to depression and work through them to make long-term changes (**Personal Agency: Responsibility**). When George relapsed shortly after discontinuing ADMs, he carried on practising MBCT-TS and said that the skills he learned were enough to pull him out of that period of low mood (**Acceptance: Perspectives on Relapse**).

**Claire**. *Claire, aged 49, had experienced four episodes of depression, beginning when she was 17. Following the MBCT-TS programme, Claire discontinued her ADMs. She relapsed and resumed medication, but subsequently tapered and discontinued again, and was not using ADMs at the time of her follow-up interview.*

At first, Claire was very sceptical about the MBCT-TS programme and thought it might all be 'mumbo jumbo'. However, she was very keen to come off ADMs, so she approached the programme with an open mind and wanted to give it her all (**Personal Agency: Control**). As the course progressed, Claire changed her mind and began to 'believe more and more that this might help me'. MBCT-TS gave her new ways to cope with her feelings, which shocked her because she 'had never took control of my depression before' (**Personal Agency: Responsibility**). She became very excited and tapered off her medication 'too quickly' and 'hit a brick wall in a short amount of time. Went straight back in to a deep depression' (**ADM Tapering/Discontinuation: Pace of Reduction**). Her doctor was very understanding, and did not push her to do anything, but advised her to go back on ADMs and try to taper off again when she was

Continued

Continued

**Box 1    Continued**

feeling better (**Interactions with GP: Presence and Support**). He said that she should try tapering them more slowly next time even though she 'wanted to get off them as soon as possible'. This time, she did 'exactly as she was told' and did not experience a relapse (**Interactions with GP: Following Advice**). Claire was very pleased because she said they had always felt that ADMs had 'suppressed' her and that the person she was when taking ADMs 'wasn't really me' (**Quality of Life: Experiencing Emotions More Fully**).

programme. Despite some of the initial reservations described earlier, participants described an open mind as key to engaging with the new psychological model, in which their thoughts, behaviours and emotions played a role in depressive relapses and recurrences: 'in the first sessions […] I switched from being highly sceptical to very interested very quickly' *(1203, Interview, Discontinued)*. Some participants articulated a move away from 'treating depression as a disease, like if you had a toothache, so you took pills', and were surprised because they 'hadn't thought that there was an alternative' *(1069, Interview, Discontinued)*. They began to feel confident to discontinue ADMs with the support of psychological therapy. In addition, people described how the programme gave them more awareness of how external factors, such as relationships or financial situations, could trigger or exacerbate depressive relapses and recurrences. On the other hand, some participants found it more difficult to engage in the programme and found themselves 'rebelling against it' because they did not have 'intellectual confidence in the process' *(3105; Written feedback, Never tapered or discontinued)*. People described how their initial treatment experiences influenced their attitudes: those who felt that the techniques were helping them to manage depressive relapse/recurrence often endorsed the psychological model. On the other hand, others who experienced deterioration in mood or relapse sometimes reported that they had reconsidered bio-medical explanations, and decided to resume or remain on ADMs: 'I really thought depression was a psychosomatic problem, but I am not so sure now. I did give it my best shot, using the mindful techniques, but I still fell into the pit of despair […] I feel that my depression is caused by a chemical imbalance in my body which, at present, is only helped by taking medication' *(2200; Written feedback, Tapered but never discontinued)*.

### Integrating models

Although some participants viewed depression as either biomedical or psychological, many did not see the two models as distinct and found ways to integrate them. For instance, they conceptualised that 'antidepressants hold onto the chemical in your body 'cause you're not making enough of it yourself', while MBCT-TS allows you to 'focus your mind onto how to make your own' *(1139, Interview, Never tapered or discontinued)*. It seemed that participants who viewed these models as compatible were more open to using ADMs and using psychological techniques as

an additional way to support their recovery, rather than viewing them as competing treatments. Furthermore, when participants observed the diversity of other people's experiences on the programme, some formed the opinion that there are 'all sorts of depressions' underpinned by different causes 'just as there are colds and flu's and viruses' *(3105, Interview, Never tapered or discontinued)*. As such, some reasoned that different people would require different treatment decisions to support recovery: 'My depression is not necessarily the same as other people's […] The right combination of changing lifestyle, specific therapies, medication whatever else it takes – that seems to be different for different people' *(3109, Interview, Never tapered or discontinued)*.

### Personal agency

This over-arching theme describes people's personal agency in their recovery and consequently their treatment choices. People described entering the study fearful about ADM discontinuation, but were hopeful that a psychological programme could support them. During MBCT-TS, people spoke about feeling better able to manage their vulnerability to depressive relapse, by using the skills and techniques they learned on the programme. While these enhanced feelings of personal agency were largely viewed as a positive and increased many people's confidence to taper and discontinue ADMs, this was not always the case. For some having more responsibility to manage their condition created a sense of unhelpful pressure. This theme comprises two sub-themes.

### Control over depression

People described how their treatment choices affected their sense of control over depression. Some felt that taking ADMs provided a sense of control, as it kept their mood on an even keel. However, this sense of control was contingent on taking ADMs, so many participants recalled how before the trial they would not consider discontinuing because they were afraid that depression would return. Through MBCT, some participants described a change in their sense of personal agency in their recovery, describing a shift from being a 'helpless victim of circumstance', to having more 'control of my feelings and my life' *(1123; Written feedback, Tapered but never discontinued)*. They reported increased awareness to recognise the early warning signs of depressive relapse and take steps to respond by applying mindfulness or cognitive-behavioural techniques from a 'toolbox', including things like meditation, activity scheduling or enlisting social support: 'Before the trial, I didn't have the tools to recognise what was happening. […] I didn't even know I was getting depressed. [Now] if things are difficult I can do something about it' *(1203, Interview, Discontinued)*. Learning these new skills reduced many people's fears about coming off ADMs, because they felt they had the capacity to prevent or contain depressive relapses. For instance, George said that before the course, he would fall into depressive episodes very suddenly and without

warning, whereas the skills learned in MBCT-TS gave him more awareness and control to act and prevent relapses before they occurred (see box 1).

## Responsibility

Participants also articulated that learning how they could have more agency over their thoughts, feelings, and behaviours led to an increased sense of responsibility to manage their well-being. Most participants viewed this as positive, especially if they were able to use the techniques to manage relapse/recurrence. Some people said they preferred MBCT-TS to taking ADMs because it made recovery feel more like a personal achievement: 'Once I've fallen and I realise that I am depressed, I take myself off and say, do 3 or 4 meditations a day. […] Which to me is better than taking a pill, because I know I've worked to get myself well' *(2016, Interview; Written feedback, Discontinued)*.

However, not all participants viewed having more agency and responsibility over their well-being positively. In particular, some participants described how this made them feel like it was their fault if they relapsed or felt they had to resume ADMs: 'I feel sad and disappointed that stopping [ADMs] made me feel low again. […] It makes me feel I'm not right in the head compared to others. I also feel annoyed with myself for not utilising MBCT skills learnt better' *(2123; Written feedback, Discontinued and resumed)*. Furthermore, a substantial number of participants expressed the challenge of finding the time, motivation or self-discipline to keep up a regular mindfulness practice outside of the group sessions. Therefore, the sense of control did not always feel stable, as it was contingent on finding time to practice and 'do it religiously, otherwise I would be fearful of it not being enough' *(2102; Written feedback, Never tapered or discontinued)*. Some were disappointed when they realised that psychological therapy was not an 'all-encompassing cure' *(1222, Interview, Discontinued and resumed)* and would involve an active and ongoing process of engagement with the techniques learned.

## Acceptance

This over-arching theme describes people's feelings of acceptance towards their history of recurrent depression and ongoing need to manage risk of relapse and recurrence. People reported feeling a sense of shame around long-term reliance on ADMs before the trial, feeling it labelled them as an ill person even if feeling well. After the trial, people described an increased sense of acceptance regarding their vulnerability to depression and an increased motivation to engage in self-care to support their recovery. This self-management included either ADMs and/or the psychological techniques for different people. This theme comprises three sub-themes.

## Resolving shame

Many participants recalled that before the trial they had felt 'inadequate' or unable to cope with life compared

with other people because of their recurrent depression treating it as a 'guilty secret' *(1123; Interview, Tapered but never discontinued)*. Taking ADMs had helped some people by reducing the symptoms and allowing them to return to feeling like a 'normal contributing person in society' *(2200; Interview, Tapered but never discontinued)*. However, others said that having to take ADMs on an ongoing basis gave them an underlying feeling that they were still 'not a well person' *(2102; Interviews, Never tapered or discontinued)*, even when the symptoms of depression were absent. For these reasons, some people recalled how before the trial, they found it difficult to name their depression, and 'couldn't even or wouldn't even admit to that' *(1031, Interview, Never tapered or discontinued)*. Through MBCT-TS, some participants described how they felt able to name their condition as depression for the first time. They discussed how meeting other people in the programme had made them realise that depression was not a negative aspect of their own self-identity, but an aspect of human experience: 'You realise it is part of the human condition rather than you' *(1128; Interview, Never tapered or discontinued)*, and it 'confirmed that I am a human, worthwhile person' *(2176; Written feedback, Discontinued and resumed)*. This led to increased feelings of acceptance towards depression, because participants experienced a shift away from viewing themselves as abnormal, to seeing depression as a more acceptable response to life's difficulties: 'Giving yourself credit […] 'cause at the end of the day […] our human brain is quite a complex thing, isn't it? […] There's nothing wrong in feeling like it' *(2140; Interview, Discontinued and resumed)*.

## Self-care

Participants described how developing more acceptance towards their condition improved their attitudes towards self-care. They said that accepting their vulnerability to depression allowed them to 'look at solutions' and that they finally had 'consent to actually do something about it' *(1031, Interview, Never tapered or discontinued)*. People described how they increasingly accepted that they needed to take care of themselves, and explained how the programme had taught them legitimate ways to do this such as using mindfulness practices: 'Previously was a mindset […] that I wasn't allowed to help myself feel better. […] Whereas this felt a way that I could do it without mollycoddling myself' *(1031, Interview, Never tapered or discontinued)*. Participants also described how the programme had reframed self-care not as something 'fluffy', but as 'practical' and a necessary part of their ongoing recovery: 'It doesn't make you any less male of course. [Chuckles] Or any less powerful' *(1203, Interview, Discontinued)*. In some cases, this new attitude towards self-care caused a shift such that people felt more acceptance towards taking ADMs: 'I don't feel any more when I take my pill every morning that there's something wrong with me' as they recognised it was important to do 'everything in my power to help myself' *(1177, Interview, Tapered but never discontinued)*. Some participants also described how

originally they had taken ADMs unwillingly, whereas after MBCT-TS they decided to take ADMs as an act of effective self-management: 'I used to hate taking them [ADMs] I accept [now] it's all about looking after yourself isn't it?' *(3103; Interview, Discontinued and resumed)*.

### Perspectives on relapse

One dimension of acceptance was people's perspectives on mood fluctuations and relapse itself. Some people favoured ADMs as an approach to relapse prevention, because it guaranteed them stability in their mood. For instance, when Greta experienced a deterioration in mood, she interpreted this as a sign that the MBCT-TS programme had been a 'failure' and she resumed taking ADMs (see box 1). However, this was not always the case, and many people described how participating in MBCT-TS changed their attitude towards relapse/recurrence. In particular, some people felt more able to accept mood fluctuations and even periods of depression. They described approaching them in a different way, 'thinking it was a phase that one was going through and sort of accepting, okay this is how you're feeling today' *(1159; Interview, Discontinued and resumed)*. Some people reported that they no longer wanted to 'blank out their negative emotions', and so did not resume ADMs, even if they relapsed: 'it's definitely helped me to realise that they [negative emotions] are a part of me as well' *(4057, Interview, Discontinued)*.

### Quality of life

People reflected on the ways in which treatment choices influenced their quality of life, specifically moving them from a place of coping, to a position where they could enjoy and appreciate their lives. This theme comprises two sub-themes.

### Experiencing emotions more fully

On reflecting on their experiences with ADMs, some participants said that while ADMs lessened their low mood, at the same time they 'dampen all other emotions', for instance, they could not feel 'blissfully happy, couldn't get angry, and in hindsight feel I was sedated' *(4057, Written feedback, Discontinued)*. In the context of depression, some people viewed this numbing effect as helpful, and reflected that while ADMs 'take away the euphoria that you would get when you've done something really, really, really good', this was 'a small price to pay really for not having the really dark times' *(2200; Interviews, Tapered but never discontinued)*. However, many people thought that this had negatively affected their quality of life, especially in cases where they found it hard to experience positive emotions. This appeared to influence people's decision to taper or discontinue ADMs, because they said that restoring their emotional range was an important part of their long-term vision of recovery: both George and Claire described this as a key motivator to discontinue their ADMs (see box 1). Indeed, people described how their emotional capacity increased after coming off

ADMs: 'I am more alive: my emotions aren't "levelled out" anymore. I can be happy, sad, angry or calm instead of just bland' *(4057, Written feedback, Discontinued)*. Despite this, some people found it a bit of a 'shock' at first, when faced with 'very extreme emotions and feelings' again *(1212; Feedback booklets, Discontinued)*. Therefore, people found it helpful that the programme taught them techniques to help manage this transition: 'I definitely used mindfulness during coming off the tablets to […] be aware what's going on inside and […] calm myself down, to have those little islands of tranquillity' *(4057, Written feedback, Discontinued)*. On the other hand, some participants said that despite not tapering or discontinuing ADMs, the programme had helped them to cultivate more positive emotions, and appeared to increase their quality of life on ADMs: 'I suppose the mindfulness in that respect has helped because […] by slowing yourself down you can […] capture some of that […] joy of life that possibly I would have lost' *(2200; Interviews, Tapered but never discontinued)*.

### From coping to enjoying life

Many people reflected that in their recovery journey they had been grateful for the periods of time where they were simply able to function. However, some participants said that the programme had helped them to move beyond that mind-set, and to develop more well-being and appreciate life: 'What has changed? I think my outlook on life, I love life, I really do […] People said to me […] before you used to skulk into the room, now you light up the room. […] I do enjoy life now, where I didn't before' *(2016, Interview, Discontinued)*. They valued the fact that the programme had an active focus on positive functioning, and encouraged them to take part in activities that brought happiness and joy into their life. Participants described this as an active process, facilitated by a sense of having more control and autonomy over making positive decisions in their life: 'I rearranged my life so that the things I do now are things that I enjoy and want to do' *(1203, Interviews, Discontinued)*; 'I am now making bigger future plans to make my life better and introducing new ventures' *(1031, Written feedback, Never tapered or discontinued)*.

### ADM tapering/discontinuation

The study's focus included people's experiences of tapering and discontinuing ADMs in the context of the MBCT-TS programme. This theme describes participants' views of what helped or hindered the process of discontinuation. It comprises two sub-themes.

### Timing

Reflecting on the right time to engage with different treatments, many participants felt that ADMs were helpful when they first became depressed: 'they got me out of my initial depression so that I could cope more with just everyday life' *(4007, Interview, Never tapered or discontinued)*. However, many did not envisage being on

ADMs indefinitely, and they described an increasing need for insight and self-management of depression as time went on. They thought that the MBCT-TS techniques required more effort, but supported a longer-term vision of recovery, to 'recognise what makes you depressed and to give you a way to cope with your depression throughout your life for the long-term, and a way that you can come off [ADMs]' *(4007, Interview, Never reduced or discontinued)*. As illustrated by this quotation, some participants viewed the MBCT-TS skills as part of a longer-term solution to ADM discontinuation, which extended beyond the 2-year follow-up period. Some participants reflected on how they thought the two treatments could be used in combination to support people at different parts of their journey, from depression through to recovery: 'I think you need that initial boost of antidepressant to perhaps get you back into a more rational level, and then once you've reached that, then bring in the MBCT, until you get back then you know, be weaned off. I can see that working very well really' *(1108, Interview, Never reduced or discontinued)*.

Participants discussed the pace at which they tapered ADMs, and how they perceived this to have influenced their outcomes. Some people who were worried about coming off ADMs shared that they exercised caution, testing out the psychological techniques for a set period and tapering slowly. They said this was helpful as it gave them time to learn to use the psychological techniques before giving up the support of their ADMs, 'by doing it slowly, you are learning those skills and you are finding out how you can use it. [Then] you can start dropping it [ADMs] at your own pace' *(1075; Interviews, Discontinued)*. In comparison, those who were keen to come off ADMs and were less fearful of the consequences described tapering more quickly. Although the programme had included explicit guidance to taper gradually, participants' reports suggested that many people had gone against this advice, and were looking for a 'quick fix' to 'get off the pills as quick as possible' *(2131, Interview, Tapered but never discontinued)*. However, on reflection many people thought, 'perhaps that wasn't the answer perhaps the thing ought to be graded on over a longer period' *(2131, Interviews, Tapered but never discontinued)*. Some of these participants reflected that in retrospect they should have been more cautious, and tapering too quickly had led to poorer outcomes: 'I reduced my tablets too quick and paid the price by having to get straight back to the full dose' *(2016, Interview; Written feedback, Discontinued)*. However, some people, like Claire, who did not successfully discontinue on their first attempt reported how they had then tried again, tapering more gradually and with more success (see box 1).

### Managing withdrawal effects
People said that the programme had helped them to cope with withdrawal effects during and after tapering/discontinuing ADMs. They described how the group and the meditation techniques provided ongoing support to manage this period: 'I used meditation techniques […]

tried to treat myself with pleasurable experiences and told myself that this would pass over. […] I had a network of fellow participants and a trustworthy instructor. All of this put me in a position of confidence that it would work this time' *(4057, Written feedback, Discontinued)*. In addition, people said that they were better able to differentiate the side effects of ADM withdrawal from a depressive relapse. For instance, Mandy said that in the past, withdrawal effects had been the biggest hindrance to tapering ADMs, because she had always mistaken them for a depressive relapse and resumed her medication. On the programme, she learned how to differentiate between these effects and 'real relapses', and said that tapering was relatively 'easy' this time around (see box 1). Indeed, some people recalled attempting to discontinue ADMs before the trial and their withdrawal symptoms being 'misdiagnosed' 'as recurring depression', whereas this time they 'knew what was coming' *(4057, Written feedback, Discontinued)*.

### Interactions with GP
Participants' described their interactions with their GPs as being important in their recovery, in both positive and negative ways. This theme comprises two sub-themes.

### Presence and support
Participants described having a GP who was easy to access throughout the process of discontinuation as supportive: 'Knowing that I could ring the doctor and say, "I need to make an appointment, I need to come and see you." There was always that net underneath me to catch me if I was falling and I couldn't stop it' *(2090, Interview, Discontinued)*, whereas some participants said they found it 'very difficult' to access their GPs, and so felt 'unsupported' *(1123, Written feedback, Tapered but never discontinued)*. Participants reported a more positive attitude to the programme if their GP had endorsed it, and some said they had only been convinced to take part in the trial because their GPs said they had themselves done a mindfulness course. When GPs encouraged their patients to use the mindfulness practices, this appeared to be associated with better engagement and subsequent success in ADM tapering and discontinuation: 'I did reach a stage where I went to see my G.P. as the depression was returning. […] We decided that I should try the exercises before trying pills. I did not need to go back on them yet […] My GP is a great help' *(2090, Written feedback, Discontinued)*.

### Following advice
Participants differed in the extent to which they sought and followed the advice of their GPs. For instance, some participants described that they remained on their ADMs at their GP's suggestion: 'My GP would not allow me to come off my antidepressant or reduce it because I had been on them so long term. [I am] relieved but also a bit disappointed' *(1108; Written feedback, Never tapered or discontinued)*. This adherence to medical advice seemed to be greater for participants who had more concerns about discontinuation. For instance, Claire, who relapsed

the first time that she had attempted tapering and discontinuation, was much more receptive to her GP's advice the second time around, because she was afraid of relapsing again (see box 1). On the other hand, where people were confident that they had learned the skills to self-manage their depression without ADMs, they more often reported that they could manage the process independently, and placed less value on their GP's advice: 'I went along to the doctors because I was polite to ask him if I could stop taking it. And he said, "Well yeah maybe in a few months time you can taper it- ease it off a bit." But really I had decided (laughs) I was going to stop. So I was just there out of politeness' *(1203; Interview, Discontinued)*.

## DISCUSSION
### Statement of principal findings
This study explored the recovery journeys of people with recurrent depression who followed a programme (MBCT-TS), designed to teach psychological skills to prevent depressive relapse while providing advice to encourage tapering and discontinuation of maintenance ADMs.[13] Thematic analysis suggested people have unique recovery journeys, but tend to be characterised by six common themes. Five illustrative stories are represented in case studies (box 1). The over-arching themes in participants' accounts were: beliefs about the causes of depression, personal agency, acceptance, quality of life, ADM tapering/discontinuation and interactions with GP (table 2). Together, these findings have the potential to facilitate discussions between clinicians and patients about the depression recovery journey. The findings also provide a starting point for more research into which treatments for recurrent depression, or combination of treatments, work best for whom and when.

### Strengths and weaknesses of this study
This study had a number of methodological strengths. We had a relatively large sample and purposively sampled the population for whom this research is relevant—people with a history of recurrent depression, stable on maintenance ADMs who were open to both a psychological and pharmacological approach to recovery. The study's time frame enabled participants to reflect on their journey with MBCT-TS and ADMs over 2 years. To support participants' recollection we developed prompts about the course of their depression and ADM use over the 2-year period based on information we had collected as part of the parent RCT.

Alongside these strengths, it is important to consider the context within which the study took place and its implications for interpretation of the findings. First, the trial was pragmatic in that it recruited participants from a particular population.[16] However, it did not include people either unwilling to consider a psychological therapy or unwilling to consider tapering/discontinuing their medication. Second, the parent trial included monitoring participants' use of ADM, and if people following MBCT-TS were not tapering/discontinuing they were invited to discuss this with their GPs. Some participants reported feeling pressured to discontinue ADMs and it is reasonable to assume that some participants may have made different decisions in a more naturalistic setting. Third, our purposive sampling means this study does not speak to a larger population of people with a history of depression who are not interested in a psychological approach and tapering/discontinuing their ADMs, or indeed prefer not to use ADM. Fourth, the questions in our feedback booklets and interviews had a particular framing, and it is possible that if the questions were framed differently the answers too may have been different. Finally, for pragmatic reasons we did not ask participants' feedback on the themes as is sometimes done in qualitative research.

### Implications of our findings
Several RCTs have demonstrated that psychological therapies such as CBT and MBCT can support ADM discontinuation,[8–10] but to our knowledge, no qualitative studies have examined people's experiences of this process. This study adds to the body of literature suggesting that people's journey involves choices among different treatments, shaped by their prior beliefs, expectations, experiences and interactions with their GPs. In both MBCT and CBT people learn new skills to manage depressive symptoms, gaining new perspectives drawn from both the psychological model and peer-to-peer learning, and develop an increased sense of agency concerning ADM discontinuation.[11 20 21] In this study participants described learning attitudes towards self-care that were participatory and empowering, which facilitated a sense of agency around ADM use, tapering and discontinuation. On the other hand, some people's biological beliefs about depression, positive experiences of ADM, and/or negative experiences of psychological therapies meant they were happy to use ADM as their primary approach to recovery.

People also emphasised the importance of a GP who is accessible and able to provide support that is collaborative, individualised and empowering, with careful monitoring over time. Moreover, they described how GPs had powerfully shared their models of depression, expectations of treatment and treatment choices. The implication for GPs is to provide accessible, collaborative, individualised and empowering care. Moreover GPs should provide people with explanatory models of depression that are bio-psycho-social alongside appropriate pharmacological and psychological treatment choices. Our findings alongside others[21–23] also suggest GPs should not offer an overly simplistic biological model, for example, 'your serotonin levels are low', followed by a (repeat) prescription of ADM.

ADMs are currently the mainstay treatment approach to recurrent depression. Kendrick has argued that many people remain on ADMs without clinical need and could benefit from support and guidance on how to discontinue, especially regarding how to deal with initial withdrawal

symptoms.[23] Our findings underscore this. Moreover, participants spoke of the importance of feeling that they had acquired from alternative skills in MBCT-TS to support their recovery generally by being able to manage their depression, but also ADM tapering and discontinuation specifically. For example, where life circumstances were challenging some people felt that the time was not right for them to discontinue ADMs; they made an informed decision to continue with their medication. Even so, the majority of participants who remained on ADMs reported that the MBCT-TS treatment had increased their quality of life on ADMs, and improved their confidence in future discontinuation when circumstances were more favourable. Our analysis also outlines participants' views on the appropriate timing of different treatments, which provide ideas for when it might be an appropriate time to initiate conversations about ADM and MBCT treatment choices.

People described how their expectations of both MBCT and ADM influenced their treatment choices. Although it is widely assumed that positive expectations predict greater benefit in psychological therapy, in our sample both unrealistically positive expectations (eg, expecting MBCT-TS to be an 'all-encompassing cure-all') and very negative expectations (having 'no intellectual confidence in the process') appeared to act as a barrier to engagement. These findings are consistent with those of Malpass *et al*[7] and suggest that openly discussing expectations at key junctures is likely to be key in preventing disappointment or disengagement from what is an effortful process of change. Likewise, in line with Maund *et al*'s findings,[6] people's causal models of depression also appeared to influence their expectations and engagement with psychological therapy. Moreover, they were subject to change during and beyond the therapy process, as their experiences either confirmed or disconfirmed their expectations and working model of depression and its treatment. People who suffer from depression frequently endorse biomedical explanations,[24] and this was evident in our sample. A number of people reported that they derived these models from discussions with their GPs as a rationale for taking ADMs. Previous research suggests that conceiving depression as a biomedical illness can absolve people of personal responsibility and thus challenge stereotypes of depression resulting from personal weakness.[7] However, our findings suggest that strongly held biomedical beliefs appeared to increase feelings of dependency on ADMs, and contribute to negative expectations and lack of engagement with psychological therapy. On the other hand, while learning a psychological model of depression empowered people towards more self-management of depression and feelings of mastery over their emotional well-being, in some cases, when people developed a psychological understanding and then went on to relapse, they blamed themselves. In some cases, practical life circumstances also made it very difficult for people to engage in an approach that required time and effort. Together, this suggests that polarised beliefs about the causes of depression can either compromise self-efficacy or promote self-blame. Many participants found it helpful to bridge biomedical and psychological theories, with parallels to a 'biopsychosocial' framework,[25] rather than viewing separate theories as competing, which seemed to foster more flexibility, self-compassion and open-mindedness towards trying different treatment options at different times in their journey of managing recurrent depression. This highlights the importance of recognising that a myriad of factors, including genetic vulnerability and challenging social circumstances, can influence depression.

People sometimes describe ADMs as sedating or numbing.[26] In these interviews participants said this could influence their decision-making about ADM use. For example, in the instance of numbing, some people viewed this as helpful as it reduced their feelings of depression, whereas other people said that ADMs numbed all of their emotions, including positive feelings, and this contributed towards a desire to discontinue them. These findings add to ongoing discussion about the psychoactive effects of ADMs, including their potential benefits and costs, how these effects impact people's experience of recovery from depression and how participating in psychological therapy can interact with these experiences.

Finally, descriptions of the role of the GP in supporting ADM discontinuation varied markedly, and this appeared to result both from differences between patients in their preferred level of guidance and support, and the availability of their GPs to provide this. For example, some people adhered to their GP's advice although this was in conflict with their own desired approach, some described informing their GP of their intentions as an act of courtesy, and some did not involve their GP at all. In some of these latter cases, participants felt that they would have benefited from more support, but their GP, for a range of reasons, was unable to provide this. People also described needing more understanding and support over time as they took more responsibility for managing their depression. This is in line with findings from Malpass *et al*,[7] who suggested that people vary in the extent to which they want to be involved in treatment decision-making, and their preferences for involvement are dynamic, not static. Archer has described different 'modes of reflexivity' noting the varying degrees to which people act autonomously or rely on endorsement from others.[27] Moreover, recovery meant different things to different people, and overall, the outcome most important to patients appeared to be their day-to-day functioning and quality of life. It is likely that when GPs are able to recognise their patients' preferred mode of engagement, and complex, dynamic views of recovery and adapt their approach accordingly, this will facilitate patient–GP consultations about ADM and psychological therapies treatment choices.

## Unanswered questions and future research
This work adds to the emerging literature on people's experiences of recovery from depression with ADM and psychological therapies. Applied research asking

how patients and health professionals communicate about their respective models of depression, and understand how this affects treatment decisions, compliance, outcomes and a broader conceptualisation of recovery would be valuable. Extending this to the broader population of people who suffer depression would not only provide an interesting and important alternate perspective but also will be important to consider with respect to recovery journeys and treatment choices.

Our work took a qualitative approach. An obvious next question asks how these process variables affect outcomes. That is to say, what works for who, how, when, to affect treatment outcomes? Finally, such research should prioritise the outcome that is most meaningful to patients: their day-to-day functioning and quality of life.

### Dissemination declaration
The trial results were disseminated in workshops and via a flyer to all participants who requested this feedback. The findings of this study will be disseminated to relevant audiences through University of Oxford communications.

**Acknowledgements** We would like to thank Trish Bartley for her input to mindfulness-based cognitive therapy (MBCT) therapist training and MBCT fidelity checks. We are grateful to members of our trial steering committee (Chris Leach, Richard Moore and Glenys Parry) and data monitoring committee (Paul Ewings, Andy Field and Joanne MacKenzie) for their valuable advice and support during the project. We acknowledge the additional support provided by the Mental Health and Primary Care Research Networks. We also acknowledge the support provided by the Department of Health and local Primary Care Trusts, in meeting the excess treatment and service support costs associated with the trial. Thanks go to the research team, who facilitated wider qualitative work in the trial, including Aaron Causley, Anna Hunt, Pooja Shah, Holly Sugg, Harry Sutton and Matthew Williams. Above all, we are grateful to the participants for their time in taking part in this trial.

**Contributors** WK, RB and NM were responsible for the PREVENT trial protocol (A randomized controlled trial comparing mindfulness-based cognitive therapy with maintenance anti-depressant treatment in the prevention of depressive relapse/recurrence) and secured the study funding. NM designed the over-arching qualitative process study to elicit service users' experiences of treatment, with input from RB, FG, RH, JC and WK. Interviews were conducted by FG and AW, supervised by NM. CC, WK, JR and AT developed the analytical strategy and protocol for the study reported here, and AT conducted the bulk of the analysis, with input from other members of the analytical team. AT, CC and WK drafted the manuscript. All other authors read the manuscript, revised it for significant intellectual content and approved the final manuscript. As chief investigator, WK had overall responsibility for the parent trial within which this study was embedded. The University of Exeter held responsibility for the parent trial and this work. WK is guarantor and corresponding author for the study.

**Funding** The PREVENT trial was funded by the National Institute for Health Research (NIHR) Health Technology Assessment (HTA) programme (08/56/01) and was published in full in Health Technology Assessment. The views and opinions expressed therein are those of the authors and do not necessarily reflect those of the HTA programme, NIHR, National Health Service or the Department of Health and Social Care. A trial steering committee, data and ethics monitoring committee, the UK Mental Health Research Network, the Primary Care Research Network, and the Comprehensive Local Research Network all provided support to the project. WK, CC and AT are supported by the Wellcome Trust (107496/Z/15/Z). RB received funding from NIHR Collaboration for Leadership in Applied Health Research and Care South West Peninsula.

**Disclaimer** The funders had no role in the design of the study, in the collection, analysis and interpretation of the data, in the writing of the report or in the decision to submit the article for publication. All authors are independent of the funders, had full access to all of the data in the study and can take responsibility for the integrity of the data and the accuracy of the data analysis.

**Competing interests** None declared.

**Patient consent for publication** Not required.

**Ethics approval** The South West Research Ethics Committee approved the trial (09/H0206/43), which was registered with the International Standard Randomised Controlled Trial Register (ISRCTN26666654) and the Medicines and Healthcare products Regulatory Agency (2009-012428-10).

**Provenance and peer review** Not commissioned; externally peer reviewed.

**Data availability statement** No data are available. We will not be making the data publicly available due to its highly confidential and identifiable nature.

**ORCID iDs**
Nicola Morant http://orcid.org/0000-0003-4022-8133
Willem Kuyken http://orcid.org/0000-0002-8596-5252

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
