## [Reviewer comments · BMJ Open]

ARTICLE DETAILS

TITLE (PROVISIONAL)	RECOVERY FROM RECURRENT DEPRESSION WITH MINDFULNESS-BASED COGNITIVE THERAPY AND ANTIDEPRESSANTS: A QUALITATIVE STUDY WITH ILLUSTRATIVE CASE STUDIES
AUTHORS	Tickell, Alice; Byng, Richard; Crane, Catherine; Gradinger, Felix; Hayes, Rachel; Robson, James; Cardy, Jessica; Weaver, Alice; Morant, Nicola; Kuyken, Willem

VERSION 1 – REVIEW

REVIEWER	Neeltje Batelaan MD PhD Department of Psychiatry Amsterdam UMC, location VUmc, Amsterdam, the Netherlands
REVIEW RETURNED	23-Sep-2019

GENERAL COMMENTS	The aim of the current paper was to describe and interpret people's experiences in the two year after taking part in a psychological programme to support discontinuation of maintenance treatment. Given the tremendous rise in long-term antidepressant use, the topic of discontinuation of medication is important, and the authors should be acknowledged for their efforts to examine patient perspectives alongside the trial. The relevance is clearly described in the introduction section, the authors elaborate on their findings and relate their findings to previous work, and the article is well written. My main concern relates to the methodology of the paper. For qualitative research, the COREQ guidelines should be used to safeguard a sound research strategy and to prevent biases to occur. The authors do not mention whether they have used this guideline and at what points the guidelines have not been followed. In qualitative research, the aim is to elicit various opinions that are not directed by the researcher. To achieve this, a diverse sample is required. The authors mention they 'purposively sampled a subgroup to represent a spread of characteristics and experiences....' (pg 10), and in their strengths and weaknesses section (p31) they mention their 'sampling approach that captured a range of perspectives' as an important strength. From this perspective, I do not understand why they only included participants who at least attended 50% of psychological therapy sessions. Excluding participants who dropped out is likely to result in a bias towards a positive experience with psychological therapy as participants with negative experiences had dropped out and could not participate. Moreover, whereas qualitative research is meant to ask open questions, the questions in the interview were not. For example, the question 'was anything from the mindfulness course useful at the time of wobbling' is likely to elicit positive
---

	experiences. By contrast, the question ‘did your use of antidepressants play a part in wobbling or starting to feel low’ specifically asks for negative experiences with medication. Authors conclude that psychological therapy can support people (abstract), and highlight the potential clinical value in providing group-based psychological interventions’. This conclusion might be biased. Although the authors acknowledge in the ‘strengths and limitations section (p6) that their sample may not be ‘fully representative’, the potential bias that may have resulted from both their inclusion criteria as well as their questions should be clearly stated. Some other methodological concerns include the following:  - Joint analysis of booklets and interviews: why was this done? What was the rationale for using booklets and doing interviews? - A coding frame was developed based on 8 interviews. After that, the lead researcher analysed 42 interviews against this coding frame. Only one researcher coding the interviews is a diversion from the COREQ guidelines. - The coding frame was adapted when themes emerged. However, qualitative research is an ongoing process, in the sense that when new themes emerge, also the interviews should be adapted. In the manuscript, it seems that interviews were all completed before coding, as a result of which new themes could not be explored further.  - In qualitative research, frequently participants who have been interviewed also provide feedback on the findings. In this study, feedback was given by researchers at international conferences. Why did the researcher assume that researchers at an international conference could provide feedback on the experiences of participants?
--	--

REVIEWER	Gregory Simon Kaiser Permanente Washington Health Research Institute USA
REVIEW RETURNED	07-Oct-2019

GENERAL COMMENTS	The topic is certainly of interest to a broad audience of primary care and mental health clinicians. The methods are appropriate to the question, and all aspects of the methods are clearly described. I have, however, some serious reservations regarding the authors’ conclusions. These methods (purposive or non-random sampling and qualitative analysis) simply cannot support any conclusions regarding effectiveness or other impact of the MBCT intervention – especially since we have no corresponding data regarding the experiences of people in the control condition. These data can certainly provide valuable information regarding how people experienced this program and can inform future intervention development and implementation. Understanding diversity of experience is quite valuable, but such data do not support causal inference. It is absolutely necessary to confine conclusions to those that can be supported by this design, these methods, and the observed data. The following specific statements should be deleted or significantly revised: Abstract: “Psychological therapy can support people with a history of recurrent depression to discontinue ADMs” “psychological therapy may increase quality of life whether or not the person successfully discontinues their ADMs”
--

	Discussion: “Our findings support the idea that psychological therapy can help to address many of the barriers and provide facilitators to tapering/discontinuation of ADMs” “We also found that engaging in psychological therapy supported learning attitudes towards self-care that were participatory and empowering, which facilitated ADM discontinuation” “Our findings support this, for example suggesting that psychological therapy can enable people to better differentiate physical and mental symptoms related to withdrawal from those related to depressive relapse.” “The present study also built on existing work highlighting the idiosyncratic ways that ADMs pharmacologically affect the mind and body through sedation, numbing, and activation.” “the benefits of psychological therapy to assist ADM tapering/discontinuation are not fully captured by rates of complete discontinuation or relapse rates of depression alone” In each of these cases, it is certainly possible to summarize and interpret the diversity of participants’ experience – especially emphasizing any new or unexpected findings. It is useful to describe areas of consensus and divergence among participants experiences and interpretations. But it is not possible to make any causal inference regarding effects of MBCT or effects of medication.
--	---

REVIEWER	Hannah Bowers University of Southampton, UK
REVIEW RETURNED	18-Oct-2019

GENERAL COMMENTS	This paper is of great value in understanding how to help manage discontinuation of antidepressant medication in patients who are at risk of relapse. This is a well-conducted piece of research. The methods and results are reported very clearly. The rationale for the study is well-presented and the implications of the findings are discussed fully. There are a few areas of this paper which may be improved through minor revision. Abstract While the abstract does cover most of the important information, the discussion of the results is somewhat limited. For example, it is unclear how ‘timing’ relates to the patient experience. Similarly, the conclusions seem to be focused on the effects of the intervention on quality of life and could be linked more clearly to the results. Introduction The aim in the introduction is slightly different from the objective in the abstract. It’s unclear if the paper is exploring the views and experiences of patients who had MBCT-TS more broadly, or is aims to understand why some patients discontinued and others did not. This could be clarified somewhat to be more consistent in the abstract and introduction.
---

	Results The results of this study are very well-explored and reported. The content covered is very important and will be valuable to the readers of this journal. There are a few areas where the reporting of the results could be improved. The managing expectations theme is a little unclear: In what way were expectations being managed or did they need to be managed? Perhaps this needs further explanation or the sub-theme renaming. This sub-theme seems to describe patient expectations, but not 'managing' expectations specifically. The relation between this and 'control' is also a little unclear. This sub-theme may benefit from further explanation or re-defining. Patients were interviewed 24 months after the intervention period. There are a number of references to patients' views before the intervention. It would be helpful to make clear that patients are reporting their earlier beliefs. The current wording gives the impression that the data directly reflects patient views from before the trial (i.e. that patients were interviewed before the trial). For example, Page 21, line 44-46 could instead say "Patients reported feeling a sense of shame around taking ADMs before the trial..." Under 'perspectives of relapse', it could be made slightly clearer how this relates to acceptance (i.e. the acceptance of risk of relapse, acceptance of mood changes etc.). The results are very detailed and clear. They cover really important findings. However views on GP interactions and the timing of withdrawal sometimes covered views that are not specific to the intervention which was trialled. It may be worth considering condensing the results section slightly by focusing on the themes and sub-themes which align more closely to the aims of the study which are specific to the use of MBCT-TS. Alternatively, it may be that revising what is written about GP interactions and timing could help keep a more narrow focus for the paper. Discussion The findings are discussed in great detail with a very informative reflection on previous findings and the implications from this study.
--	--

REVIEWER	Harm Van Marwijk Brighton and Sussex Medical School UK
REVIEW RETURNED	20-Oct-2019

GENERAL COMMENTS	This paper is a qualitative spinoff of a randomised trial. Reading it, I felt that a proper mixed-methods approach would perhaps have been more interesting than what they now did. Properly combining both methods would have yielded more innovative findings perhaps. The focus in this project seems to have been more on the RCT than the qualitative work. With a negative trial, that makes it a little hard to make sense of these data for the general reader. I miss a lot of our work around recurrence and particularly the recent Bosman et al. paper in the BJGP. That makes me wonder how thorough the embedding in the literature was. The relationship between the results and the literature is not strong anyway. A meta-synthesis (or a reference to one), would have been good. The critical question seems off the mark, somehow: how participants in this trial described the impact of MBCT-TS on their patterns of ADM usage over a 24-month follow-up period, to try to understand why some people, but not others, discontinued. That is quite a particular context. Why would that inform a wider audience? As a practising GP, the themes are hardly surprising.
--

	As a researcher in this field, the results are even less surprising. The paper is also very long. There is a list of experienced coauthors on the paper. I would suggest they help to try to write a new version of 2500 words max (is there no word count for this journal?). A coauthor could also perhaps work to triangulate the data: I feel some nuance may have been missed. Another thing that interests me is that the bulk of antidepressants is prescribed in primary care. For instance: should we as GPs be more careful with our 'biological' explanations for depression? Should it be less comfortable to prescribe antidepressants? The interesting fact here is that their overall effectiveness for a first episode is limited, but their effects might be better for recurrence prevention. To give another example of a lack of focus: The abstract reads: 'Conclusions: Psychological therapy can support people with a history of recurrent depression to discontinue ADMs by teaching skills to manage depressive symptoms and the tapering process.' Is that something the authors find? Or a more general statement? What is the relationship with their work? 'However, this is an effortful process, requiring time and motivation to learn and apply psychological techniques, relatively stable life circumstances, and adequate support from medical professionals.' How new is this sentence for the reader? 'Nevertheless, psychological therapy may increase the quality of life whether or not the person successfully discontinues their ADMs or experiences further depressive symptoms.' These seem open doors. Could become a nice paper but still needs a lot of work.
--	---

VERSION 1 – AUTHOR RESPONSE

Reviewer: 1

Reviewer Name: Neeltje Batelaan MD PhD

Institution and Country: Department of Psychiatry Amsterdam UMC, location VUmc, Amsterdam, the Netherlands

Please state any competing interests or state 'None declared': None declared

Please leave your comments for the authors below

The aim of the current paper was to describe and interpret people's experiences in the two year after taking part in a psychological programme to support discontinuation of maintenance treatment. Given the tremendous rise in long-term antidepressant use, the topic of discontinuation of medication is important, and the authors should be acknowledged for their efforts to examine patient perspectives alongside the trial. The relevance is clearly described in the introduction section, the authors elaborate on their findings and relate their findings to previous work, and the article is well written. *We thank Dr Batelaan for her encouraging comments about the focus and potential importance of our paper.*

My main concern relates to the methodology of the paper. For qualitative research, the COREQ guidelines should be used to safeguard a sound research strategy and to prevent biases to occur. The authors do not mention whether they have used this guideline and at what points the guidelines have not been followed.

The BMJ uses its checklist, based on SRQR guidelines (not dissimilar to those suggested by Reviewer 1). We reference the BMJ checklist for qualitative work as well as the COREQ guidelines, which are included in the supplementary materials.

In qualitative research, the aim is to elicit various opinions that are not directed by the researcher. To achieve this, a diverse sample is required. The authors mention they 'purposively sampled a subgroup to represent a spread of characteristics and experiences....' (pg 10), and in their strengths and weaknesses section (p31) they mention their 'sampling approach that captured a range of perspectives' as an important strength. From this perspective, I do not understand why they only included participants who at least attended 50% of psychological therapy sessions. Excluding participants who dropped out is likely to result in a bias towards a positive experience with psychological therapy as participants with negative experiences had dropped out and could not participate.

The paper focuses on a particular population, namely people at risk of depressive relapse who had been on m-ADM and were open to a psychological approach (MBCT) as a way of learning to stay well and potentially taper and discontinue their ADM. We therefore sampled people who had participated in the programme and had a range of recovery journeys with respect to experiences of treatment and outcomes. In this sense our recruitment was purposive. We have made this clearer in the methods and interpretation.

Moreover, whereas qualitative research is meant to ask open questions, the questions in the interview were not. For example, the question 'was anything from the mindfulness course useful at the time of wobbling' is likely to elicit positive experiences. By contrast, the question 'did your use of antidepressants play a part in wobbling or starting to feel low' specifically asks for negative experiences with medication. Authors conclude that psychological therapy can support people (abstract), and highlight the potential clinical value in providing group-based psychological interventions'. This conclusion might be biased. Although the authors acknowledge in the 'strengths and limitations section (p6) that their sample may not be 'fully representative', the potential bias that may have resulted from both their inclusion criteria as well as their questions should be clearly stated. *We chose to retain the same interview schedule across all participants to ensure meaningful comparisons were possible. However, in accordance with standard semi-structured interview design enough flexibility was built into each individual interview to enable the interviewer to respond to participants' needs and concerns as they emerged. We also piloted the schedule and involved people with lived experience of depression in its content and in the training of the researchers who conducted the interviews. The Method section includes the following text "Interviews were semi-structured and normally conducted face-to-face by trained researchers, approximately 24 months after MBCT-TS. They lasted between 45 minutes and one hour and explored experiences during the follow-up period, with questions addressing times of wellness, early signs of potential depressive relapse, and relapses. Questions explored the use and perceived value of mindfulness techniques, use of ADMs, and their combination. We tailored interviews to the specific profile of each participant using a 'timeline' prepared in advance and amended by the participant at the interview, which summarised each participant's ADM use, relapses, and significant life events, as reported to the research team during the trial. The interview schedule was deliberately broad in focus and is provided in the online supplementary materials."*

However, we acknowledge that the questions were framed with the particular population of people and research focus in mind, and this will have introduced bias. We have tried to make this framing clear in the introduction and methods and acknowledge in the study limitations that a different approach might have yielded somewhat different themes. However, we could note that these are broadly in line with related qualitative work and our clinical experience of working with this population supporting them with their recovery.

We agree that the abstract and conclusions needed to be more grounded in the findings and these have been substantively redrafted. The interview schedule was extensively piloted with people with a history of depression and exposure to both MBCT and m-ADM and we have confidence in its ability to generate data that answers the research question. We have responded to the issue about the inclusion criteria above.

Some other methodological concerns include the following:

- Joint analysis of booklets and interviews: why was this done? What was the rationale for using booklets and doing interviews?

We used these two approaches very deliberately. The interviews were more in depth but took place two years after study entry and so the booklets, collected at two time points, provided participants more top level experiences. Finally, a strength of our work was researchers using all that we knew about participants' experiences of depression, treatment choices and earlier answers in the booklets to inform the interviews. This enabled a much richer discussion about participants recovery journey with ADM and MBCT, which is reflected in the case studies.

- A coding frame was developed based on 8 interviews. After that, the lead researcher analysed 42 interviews against this coding frame. Only one researcher coding the interviews is a diversion from the COREQ guidelines.

- The coding frame was adapted when themes emerged. However, qualitative research is an ongoing process, in the sense that when new themes emerge, also the interviews should be adapted. In the manuscript, it seems that interviews were all completed before coding, as a result of which new themes could not be explored further.

We acknowledge that although this is the approach taken in particular qualitative approaches such as grounded theory, it is not universally the approach taken in qualitative research. Our methodological approach to thematic analysis involved a team approach to the early stages of reviewing transcripts, then double-rating a subset of transcripts until the team were confident that we had a set of stable themes and the rater was sufficiently familiarised and supported to complete the analysis. Moreover, our overall N is relatively large for a qualitative study and the approach was pragmatic. We have used the BMJ guidelines for qualitative research to flag the page numbers in the manuscript where all the relevant issues are covered.

- In qualitative research, frequently participants who have been interviewed also provide feedback on the findings. In this study, feedback was given by researchers at international conferences. Why did the researcher assume that researchers at an international conference could provide feedback on the experiences of participants?

The timelag between when the interviews were done and the analysis made it impractical for us to seek participant feedback on these particular analyses. However, as described in the method the whole study had patient and public involvement running throughout it, which included patients training researchers and being involved in the research management meetings. The feedback we elicited from researchers and clinicians familiar with psychological therapies generally, MBCT specifically and m-ADM was one of the emergent themes. We have made this clearer in the manuscript and noted this in the limitations / future research section of the discussion.

Reviewer: 2

Reviewer Name: Gregory Simon

Institution and Country:

Kaiser Permanente Washington Health Research Institute

USA

Please state any competing interests or state 'None declared': None

Please leave your comments for the authors below

The topic is certainly of interest to a broad audience of primary care and mental health clinicians. The methods are appropriate to the question, and all aspects of the methods are clearly described.

We thank Dr Gregory for his positive comments about the focus and potential importance of this work,

I have, however, some serious reservations regarding the authors' conclusions. These methods (purposive or non-random sampling and qualitative analysis) simply cannot support any conclusions regarding effectiveness or other impact of the MBCT intervention – especially since we have no corresponding data regarding the experiences of people in the control condition. These data can certainly provide valuable information regarding how people experienced this program and can inform future intervention development and implementation. Understanding diversity of experience is quite valuable, but such data do not support causal inference. It is absolutely necessary to confine conclusions to those that can be supported by this design, these methods, and the observed data. The following specific statements should be deleted or significantly revised:

We acknowledge this legitimate concern and have made changes that limit our conclusions to the specific experiences as described by our groups of participants with all the limitations to generalisability that that entails.

Abstract:

“Psychological therapy can support people with a history of recurrent depression to discontinue ADMs”

“psychological therapy may increase quality of life whether or not the person successfully discontinues their ADMs”

We have deleted these statements.

Discussion:

“Our findings support the idea that psychological therapy can help to address many of the barriers and provide facilitators to tapering/discontinuation of ADMs”

“We also found that engaging in psychological therapy supported learning attitudes towards self-care that were participatory and empowering, which facilitated ADM discontinuation”

“Our findings support this, for example suggesting that psychological therapy can enable people to better differentiate physical and mental symptoms related to withdrawal from those related to depressive relapse.”

“The present study also built on existing work highlighting the idiosyncratic ways that ADMs pharmacologically affect the mind and body through sedation, numbing, and activation.”

“the benefits of psychological therapy to assist ADM tapering/discontinuation are not fully captured by rates of complete discontinuation or relapse rates of depression alone”

In each of these cases, it is certainly possible to summarize and interpret the diversity of participants' experience – especially emphasizing any new or unexpected findings. It is useful to describe areas of consensus and divergence among participants experiences and interpretations. But it is not possible to make any causal inference regarding effects of MBCT or effects of medication.

We thank Dr Gregory and we have both removed these specific instances and gone through to ensure we primarily describe people's experiences rather than tell a causal story.

Reviewer: 3

Reviewer Name: Hannah Bowers

Institution and Country: University of Southampton, UK

Please state any competing interests or state 'None declared': None

Please leave your comments for the authors below

This paper is of great value in understanding how to help manage discontinuation of antidepressant medication in patients who are at risk of relapse. This is a well-conducted piece of research. The methods and results are reported very clearly. The rationale for the study is well-presented and the implications of the findings are discussed fully. There are a few areas of this paper which may be improved through minor revision.

We thank Dr Bowers for her positive comments about our work.

Abstract

While the abstract does cover most of the important information, the discussion of the results is somewhat limited. For example, it is unclear how 'timing' relates to the patient experience. Similarly, the conclusions seem to be focused on the effects of the intervention on quality of life and could be linked more clearly to the results.

As noted above, we have reworked the abstract.

Introduction

The aim in the introduction is slightly different from the objective in the abstract. It's unclear if the paper is exploring the views and experiences of patients who had MBCT-TS more broadly, or is aims to understand why some patients discontinued and others did not. This could be clarified somewhat to be more consistent in the abstract and introduction.

Thank you, as noted above we have refocused the paper on what we believe is most important, novel and impactful within our data and ensured the introduction sets this up.

Results

The results of this study are very well-explored and reported. The content covered is very important and will be valuable to the readers of this journal. There are a few areas where the reporting of the results could be improved.

The managing expectations theme is a little unclear: In what way were expectations being managed or did they need to be managed? Perhaps this needs further explanation or the sub-theme renaming. This sub-theme seems to describe patient expectations, but not 'managing' expectations specifically. The relation between this and 'control' is also a little unclear. This sub-theme may benefit from further explanation or re-defining.

We have renamed this theme as Dr Bowers suggests and simplified the text.

Patients were interviewed 24 months after the intervention period. There are a number of references to patients' views before the intervention. It would be helpful to make clear that patients are reporting their earlier beliefs. The current wording gives the impression that the data directly reflects patient views from before the trial (i.e. that patients were interviewed before the trial). For example, Page 21, line 44-46 could instead say "Patients reported feeling a sense of shame around taking ADMs before the trial..."

We agree, have made this particular change and noted several other instances where the same changes were required.

Under 'perspectives of relapse', it could be made slightly clearer how this relates to acceptance (i.e. the acceptance of risk of relapse, acceptance of mood changes etc.).

Thank you – we have made this change.

The results are very detailed and clear. They cover really important findings. However views on GP interactions and the timing of withdrawal sometimes covered views that are not specific to the intervention which was trialled. It may be worth considering condensing the results section slightly by focusing on the themes and sub-themes which align more closely to the aims of the study which are specific to the use of MBCT-TS. Alternatively, it may be that revising what is written about GP interactions and timing could help keep a more narrow focus for the paper.

We have both focused the manuscript as outlined above, but what was clear is that participants' experiences of their GPs were a pivotal part of their recovery. Given the implications of this work for GPs and how they support people's recovery journey we have elected to retain the discussion of GP interactions.

Discussion

The findings are discussed in great detail with a very informative reflection on previous findings and the implications from this study.

We thank Dr Bowers for her encouraging feedback.

Reviewer: 4

Reviewer Name: Harm Van Marwijk

Institution and Country:

Brighton and Sussex Medical School

UK

Please state any competing interests or state 'None declared': None declared

Please leave your comments for the authors below

This paper is a qualitative spinoff of a randomised trial. Reading it, I felt that a proper mixed-methods approach would perhaps have been more interesting than what they now did. Properly combining both methods would have yielded more innovative findings perhaps. The focus in this project seems to have been more on the RCT than the qualitative work. With a negative trial, that makes it a little hard to make sense of these data for the general reader.

The study is not a "spin off." As outlined in the protocol, it was always conceptualized as a qualitative study embedded with a larger trial.

We have outlined in the introduction how the findings of the larger trial and a subsequent IPD meta-analysis suggest not only that MBCT is effective but also that is potentially an alternative to m-ADM for the many millions of people who take m-ADM and would like an alternative. We have refocused this work to describe the recovery journey of people with m-ADM and MBCT, which is a novel and important finding. We have removed all causal statements about effectiveness so that there is a clear demarcation between the parent RCT study and IPD meta-analysis which speak to effectiveness and this study which speaks to participants' experiences.

We agree that mixed approaches would yield interesting findings and this is now included at the end of the discussion as future research.

I miss a lot of our work around recurrence and particularly the recent Bosman et al. paper in the BJGP. That makes me wonder how thorough the embedding in the literature was. The relationship between the results and the literature is not strong anyway. A meta-synthesis (or a reference to one), would have been good.

We thank Dr Van Marwijk for pointing us to the very interesting Bosman paper which we now cite and which underscores the need for this work as well resonating with several of the emergent themes in our work (GP support, expectations sense of control/agency). This alongside the Malpass (2009) meta-ethnography now embed our study in the larger literature.

The critical question seems off the mark, somehow: how participants in this trial described the impact of MBCT-TS on their patterns of ADM usage over a 24-month follow-up period, to try to understand why some people, but not others, discontinued. That is quite a particular context. Why would that inform a wider audience? As a practising GP, the themes are hardly surprising. As a researcher in this field, the results are even less surprising.

As noted above we have refocused the paper and the abstract and main paper elements are now more coherently inter-related. The paper now focuses on a particular population, people on m-ADM open to psychological approaches and wishing to taper/discontinue their m-ADM.

We agree with Dr Van Marwijk that in some senses the themes are unsurprising. However, many people would like alternatives to m-ADM, MBCT is an effective psychological approach and there is therefore a need to understand people's perspectives on MBCT alongside m-ADM, so they can best be supported in their recovery journey. We disagree with Dr Van Marwijk however in that if these themes were dominant in GPs' understanding, we would not have the rates of ADM prescribing, the prevailing biological models of depression as "a serotonin deficiency" and lack of access to psychological therapies that we do. We have made this point in the discussion.

The paper is also very long. There is a list of experienced coauthors on the paper. I would suggest they help to try to write a new version of 2500 words max (is there no word count for this journal?).

As suggested we have shortened the paper and ensured we follow BMJ Open guidelines.

A coauthor could also perhaps work to triangulate the data: I feel some nuance may have been missed.

The co-authors have revisited the methods and results and the refocused paper is the result of this work.

Another thing that interests me is that the bulk of antidepressants is prescribed in primary care. For instance: should we as GPs be more careful with our 'biological' explanations for depression? Should it be less comfortable to prescribe antidepressants? The interesting fact here is that their overall effectiveness for a first episode is limited, but their effects might be better for recurrence prevention.

We agree, and the implications draw out some of the themes as they are relevant for people on m-ADM and prescribing GPs.

To give another example of a lack of focus:

The abstract reads:

'Conclusions: Psychological therapy can support people with a history of recurrent depression to discontinue ADMs by teaching skills to manage depressive symptoms and the tapering process.' Is that something the authors find? Or a more general statement? What is the relationship with their work?

'However, this is an effortful process, requiring time and motivation to learn and apply psychological techniques, relatively stable life circumstances, and adequate support from medical professionals.' How new is this sentence for the reader?

'Nevertheless, psychological therapy may increase the quality of life whether or not the person successfully discontinues their ADMs or experiences further depressive symptoms.' These seem open doors.

We agree and have rewritten the abstract and replaced much of this text.

Could become a nice paper but still needs a lot of work.

We thank Dr Van Marwijk for pointing us to a key paper and providing a constructive GP and primary care research perspective.

VERSION 2 – REVIEW

REVIEWER	Gregory Simon Kaiser Permanente Washington Health Research Institute Seattle, WA, USA
REVIEW RETURNED	10-Jan-2020

GENERAL COMMENTS	All of my concerns have been adequately addressed.
--

REVIEWER	Harm Van Marwijk Brighton and Sussex medical school UK
REVIEW RETURNED	16-Jan-2020

GENERAL COMMENTS	Well done, my compliments, great rebuttal and very nice paper now!!!
--